# TREE REWARD-ALIGNED SEARCH FOR TREASURE IN MASKED DIFFUSION LANGUAGE MODELS

## ABSTRACT

Tree search has recently emerged as a powerful framework for aligning generative models with task-specific rewards at test time. Applying tree search to Masked Diffusion Language Models, however, introduces two key challenges: (i) parallel unmasking yields highly correlated branches, limiting exploration, and (ii) reward evaluation via sampled completions produces high-variance estimates, making pruning unstable. We propose TREASURE, a tree-search test-time alignment method that addresses these issues. It introduces (i) UNMASKBRANCH, a branching strategy based on first-hitting unmasking that diversifies both token content and reveal order with a single model call per parent node, and (ii) RESUBSTITUTESCORE, a pruning rule that uses deterministic resubstitution to score partially masked sequences with low-variance proxy completions. Theoretically, we quantify branching efficiency gains in NFEs (number of function evaluations), show that the scoring rule approximates the true reward with error bounded by predictive uncertainty, and prove improvements with larger tree widths. Empirically, TREASURE achieves state-of-the-art results on perplexity, linguistic acceptability, and control of sentiment and toxicity, outperforming prior methods under matched compute budgets, with especially strong gains in low-NFE regimes.

## 1 INTRODUCTION

Masked Diffusion Language Models (MDLMs) (Nie et al., 2025; Sahoo et al., 2024; Shi et al., 2024; Yang et al., 2025b) have emerged as a compelling alternative to autoregressive models (Brown et al., 2020; Radford et al., 2019; Touvron et al., 2023). They start with all-mask tokens and gradually reveal tokens through a sequence of discrete denoising steps. At each step, the model predicts token distributions for masked positions, conditioned on the current partially masked sequence and the diffusion timestep. This formulation enables flexible sampling schedules and broad conditioning patterns, making MDLMs well-suited for controllable generation tasks.

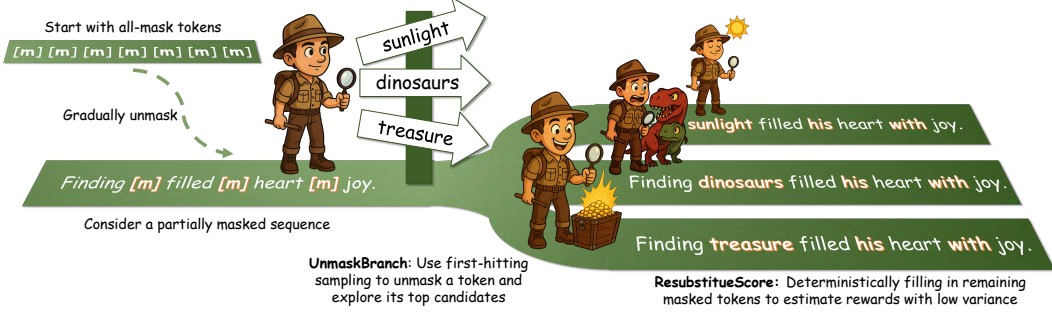

Figure 1: **Conceptual illustration of TREASURE.** UNMASKBRANCH uses first-hitting sampling to branch by selecting both which position to unmask next and which token to place there, thereby diversifying exploration. RESUBSTITUTESCORE deterministically fills in the remaining mask tokens to obtain low-variance reward estimates for pruning.

However, this flexibility is not fully realized without mechanisms to align the model's outputs with user-defined objectives. Test-Time Alignment (TTA) enables guiding language model outputs toward task-specific goals without retraining. In applications such as toxicity avoidance (Logacheva et al., 2022), sentiment control (Barbieri et al., 2020), or enforcing linguistic acceptability (Warstadt et al., 2019), aligning generation with external reward functions at test time offers a flexible and training-free alternative to supervised fine-tuning. While TTA has been actively explored in autoregressive (Liu et al., 2024; Lu et al., 2022; Ziegler et al., 2019) and continuous diffusion models (Guo et al., 2025b; Li et al., 2025; Singhal et al., 2025; Uehara et al., 2024a;b), its application to MDLMs remains limited. To our knowledge, only a few recent works have attempted to integrate reward signals into MDLM decoding at test time. For example, Singhal et al. (2025) propose Feynman–Kac steering, but their approach assumes continuous-state dynamics that may not translate well to discrete, token-level masked diffusion. Pani et al. (2025) introduce a sequential Monte Carlo method, but their evaluation is limited to image generation tasks. See Appendix A for a broader discussion.

Tree search has recently shown strong performance in aligning continuous diffusion models (Guo et al., 2025b; Li et al., 2025) at test time, offering a principled framework for balancing exploration and exploitation. However, applying tree search to MDLMs poses unique challenges. First, branching is ineffective under parallel unmasking: naïve updates often yield negligible changes, producing highly correlated trajectories and poor exploration. Second, pruning is unstable in discrete spaces: unlike continuous diffusion, where smooth latent dynamics enable reliable intermediate value estimates, MDLMs output categorical distributions per masked position, making sampled reward estimates high-variance and brittle to small logit perturbations. Overcoming these challenges requires rethinking both branching and pruning.

We propose TREASURE (**T**ree **Re**ward-**A**ligned **S**earch with **U**nmasking and **Re**substitution), a tree-search method designed for MDLMs. TREASURE introduces a branching rule based on *first-hitting unmasking*, which expands the search only at commitment events, preserving efficiency while diversifying unmasking order and token content. For pruning, it employs *resubstitution scoring*, which deterministically fills masked positions to provide low-variance reward estimates with minimal model calls. A conceptual illustration can be found in Figure 1. Theoretically, we quantify its efficiency gains in NFEs, show that its scoring rule approximates the true reward with error bounded by the model's predictive uncertainty, and establish provable improvements with larger tree widths. Empirically, across controllable generation tasks (perplexity, linguistic acceptability, toxicity, and sentiment), TREASURE achieves state-of-the-art rewards under matched compute budgets, outperforming naïve sampling, Best-of-$N$, and Feynman–Kac steering.

**Contributions.** Our contributions can be summarized as follows: **(i)** A new perspective on TTA for MDLMs via tree search; **(ii)** a branching rule that exploits unmasking events for efficient, diverse exploration; **(iii)** a pruning rule based on deterministic resubstitution for low-variance reward estimation under fixed NFE; **(iv)** theoretical guarantees including branching efficiency, pruning accuracy, and reward gains; and **(v)** state-of-the-art performance across extensive controllable generation benchmarks under a fixed compute budget.

**Notation.** Let $\mathcal{V}$ be the set of one-hot vectors in $\mathbb{R}^V$, with the $V$th component reserved for the mask token $\boldsymbol{m}$. Discrete variables are denoted by $\boldsymbol{z}_t, \boldsymbol{z}_n, \boldsymbol{x} \in \mathcal{V}$, where subscripts $t$ and $n$ indicate time and the number of masked tokens. We write $\boldsymbol{x} \sim \mathrm{Cat}(\boldsymbol{x}; \boldsymbol{p})$ if $\boldsymbol{x}$ is drawn from a categorical distribution with parameter $\boldsymbol{p} \in \Delta^V$, the probability simplex. For length-$L$ sequences, we write $\boldsymbol{z}_t^{1:L}, \boldsymbol{z}_n^{1:L}, \boldsymbol{x}^{1:L} \in \mathcal{V}^L$, with $\boldsymbol{z}_t^\ell, \boldsymbol{z}_n^\ell, \boldsymbol{x}^\ell$ denoting the $\ell$th token. Finally, $\mathrm{TopK}_b(\boldsymbol{\mu})$ returns the indices of the top $b$ entries of $\boldsymbol{\mu} \in \Delta^V$.

## 2 BACKGROUND

### 2.1 MASKED DIFFUSION LANGUAGE MODELS

We provide the necessary MDLM background here, with further details in Appendix B.

**Forward and reverse processes.** MDLMs (Sahoo et al., 2024; Shi et al., 2024) define a forward process that mixes data with the absorbing mask token $\boldsymbol{m} = (0, \dots, 0, 1) \in \Delta^V$:

$$q(\boldsymbol{z}_t | \boldsymbol{x}) = \mathrm{Cat}(\boldsymbol{z}_t; \alpha_t \boldsymbol{x} + (1 - \alpha_t) \boldsymbol{m}), \tag{1}$$

where $\alpha_t$ decreases monotonically from $\alpha_0 \approx 1$ to $\alpha_1 \approx 0$. Once a token becomes masked at time $s$, it remains so for all $t > s$, i.e., $q(\boldsymbol{z}_t | \boldsymbol{z}_s = \boldsymbol{m}) = \mathrm{Cat}(\boldsymbol{z}_t; \boldsymbol{m})$. Applied to a sequence, this corruption acts independently across positions, so tokens evolve in parallel. Learning the reverse process thus amounts to iterative unmasking. Conditioned on $\boldsymbol{x}$, the time reversal of the forward process for $s < t$ is

$$q(\boldsymbol{z}_s | \boldsymbol{z}_t, \boldsymbol{x}) = \begin{cases} \mathrm{Cat}(\boldsymbol{z}_s; \boldsymbol{z}_t), & \boldsymbol{z}_t \neq \boldsymbol{m}, \\ \mathrm{Cat}\Big(\boldsymbol{z}_s; \dfrac{(1 - \alpha_s)\boldsymbol{m} + (\alpha_s - \alpha_t)\boldsymbol{x}}{1 - \alpha_t}\Big), & \boldsymbol{z}_t = \boldsymbol{m}. \end{cases} \tag{2}$$

At inference, since $\boldsymbol{x}$ is unknown, we approximate it with a learned network $\boldsymbol{x}_\theta(\boldsymbol{z}_t, t)$, trained with the objective described in Appendix B. The learned time reversal is then $p_\theta(\boldsymbol{z}_s | \boldsymbol{z}_t) = q(\boldsymbol{z}_s | \boldsymbol{z}_t, \boldsymbol{x}_\theta(\boldsymbol{z}_t, t))$. The network prediction $\boldsymbol{x}_\theta$ is usually constrained so that (i) it assigns zero probability to the mask token, and (ii) it directly copies already unmasked tokens, ensuring that only masked tokens need to be reconstructed.

**Naïve parallel sampling.** A simple way to sample from MDLMs is to unmask positions parallelly at each step. For sequences of length $L$, this corresponds to a factorized reverse transition:

$$p_\theta(\boldsymbol{z}_s^{1:L} | \boldsymbol{z}_t^{1:L}) := \prod_{\ell=1}^{L} p_\theta(\boldsymbol{z}_s^\ell | \boldsymbol{z}_t^{1:L}) = \prod_{\ell=1}^{L} q(\boldsymbol{z}_s^\ell | \boldsymbol{z}_t^{1:L}, \boldsymbol{x}_\theta(\boldsymbol{z}_t^{1:L}, t)). \tag{3}$$

Due to the small changes between adjacent diffusion steps, only a small subset of entries are effectively unmasked at each step, leading to inefficient use of computation.

**First hitting sampling (FHS).** To mitigate the inefficiency of naïve parallel sampling, Zheng et al. (2024) proposed first hitting sampling (FHS), which only simulates the *moments when actual unmasking events occur*. Set $\tau_L = 1$. When $n$ masked tokens remain, a uniform random variable $u_n \sim \mathcal{U}(0, 1)$ is used to sample the next event time

$$\tau_{n-1} = \alpha^{-1}\big(1 - u_n^{1/n}(1 - \alpha_{\tau_n})\big), \tag{4}$$

where $\alpha^{-1}$ is the inverse noise schedule and $\tau_n$ is the current time. One of the masked positions is then chosen uniformly at random and unmasked according to the model's categorical prediction at time $\tau_{n-1}$. See Algorithm 1 in Section 3.2 for the pseudocode.

## 2.2 TEST-TIME ALIGNMENT

A common objective in TTA is to obtain generations that remain consistent with the pretrained model while maximizing a task-specific reward $r: \mathcal{V}^L \to \mathbb{R}$. This can be expressed as a KL-regularized optimization problem with a closed-form solution (Faria & Smith, 2025; Uehara et al., 2024a):

$$p_{\mathrm{tar}} = \arg\max_p \mathbb{E}_{\boldsymbol{x}^{1:L} \sim p}[r(\boldsymbol{x}^{1:L})] - \lambda D_{\mathrm{KL}}(p \| p_{\mathrm{pre}}) \propto p_{\mathrm{pre}}(\boldsymbol{x}^{1:L}) \cdot \exp\Big(\frac{r(\boldsymbol{x}^{1:L})}{\lambda}\Big). \tag{5}$$

where $\lambda > 0$ is a temperature parameter that controls the trade-off between reward maximization and staying close to the pretrained distribution. Following the entropy-regularized inference-time alignment framework (Pani et al., 2025; Singhal et al., 2025; Uehara et al., 2024b), the corresponding *soft value function* at step $t$ is defined as

$$v_t(\boldsymbol{z}_t^{1:L}) := \lambda \log \mathbb{E}_{\boldsymbol{x}^{1:L} \sim p_{\mathrm{pre}}(\cdot | \boldsymbol{z}_t^{1:L})}\Big[\exp\Big(\frac{r(\boldsymbol{x}^{1:L})}{\lambda}\Big)\Big] \approx r\big(\mathbb{E}_{\boldsymbol{x}^{1:L} \sim p_{\mathrm{pre}}(\cdot | \boldsymbol{z}_t^{1:L})}[\boldsymbol{x}^{1:L}]\big). \tag{6}$$

The last approximation involves two simplifications: replacing the log–exp expectation with a direct expectation, and evaluating the reward at the mean instead of averaging. This reduces the problem to approximating the soft value function at intermediate steps rather than computing it exactly.

## 3 METHOD

We introduce TREASURE, a TTA method for MDLMs based on tree search. Classical tree search consists of two components: (i) branching, which expands the search frontier by generating diverse candidate continuations, and (ii) pruning, which retains only the most promising nodes using a task-specific value function. We first show how this framework applies naturally to *continuous* diffusion models, then highlight the unique challenges in MDLMs, and describe how TREASURE rethinks branching and pruning to address them.

### 3.1 RETHINKING BRANCHING AND PRUNING FOR MDLMS

**Tree search for continuous diffusion models.** In continuous diffusion models, tree search is a natural fit (Li et al., 2025). Each reverse step $p_\theta(z_{t-1}|z_t)$ is stochastic (e.g., in DDPM (Ho et al., 2020)), so branching arises naturally by sampling multiple candidates for $z_{t-1}$. Because the latent space is smooth, the model's prediction $\hat{x}_0(z_t, t)$ (or short rollouts) yields reliable intermediate reward estimates $r(\hat{x}_0(z_t, t))$, enabling effective pruning by discarding low-value branches.

**Challenges for branching in MDLMs.** The key challenge is that branching in MDLMs does not naturally produce diverse trajectories. This is due to two structural properties of masked diffusion. First, sampling is performed *in parallel*: all masked positions are updated simultaneously, resulting in tightly coupled token distributions. Repeated sampling from the same state yields highly correlated or nearly identical candidates, so naive branching by repeating the sampling multiple times explores only a narrow subset of the space, as illustrated in Fig. 2. Second, the *unmasking schedule* which determines which tokens are revealed at each step is decided endogenously by the model (e.g., via confidence thresholds). Since this schedule is unpredictable, local resampling rarely alters which tokens are committed next, limiting tra-

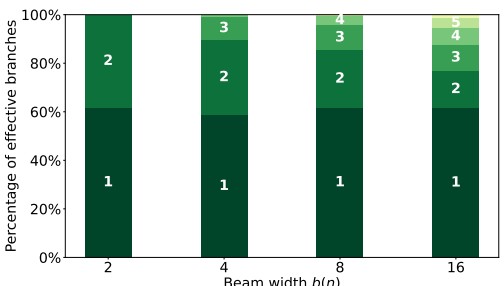

Figure 2: **Distribution of effective branch counts under different beam widths** $b(n)$. Despite wider beams, most nodes collapse to only one or two effective (distinct) branches, indicating that parallel sampling produces many redundant branches.

jectory diversity. These phenomena make straightforward token-level branching ineffective, requiring a rethinking of how to construct and expand the search tree.

**Challenges for pruning in MDLMs.** Unlike continuous diffusion models, which predict a point in latent space, MDLMs output a *distribution* over vocabulary tokens at each masked position. Estimating the value function thus requires sampling a complete sequence from this distribution,

$$\hat{v}_t(\cdot) := r\big(\hat{x}_0^{1:L}(z_t^{1:L})\big), \qquad (7)$$

where $\hat{x}_0^{1:L}$ denotes a random sample from $x_\theta(z_t^{1:L}, t)$ (Singhal et al., 2025). This introduces high variance: small perturbations in $\hat{x}_0^{1:L}$ can cause large changes in the resulting reward. In principle, this variance could be reduced by sampling multiple completions per node, but doing so greatly increases computa-

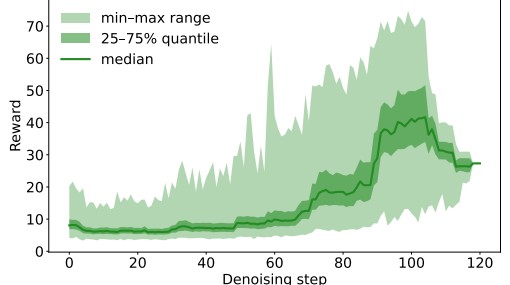

Figure 3: **Reward variation across denoising steps on CoLA.** Median (line), interquartile range (25–75%, dark green), and min–max range (light green) reveal large fluctuations, underscoring the need for stable pruning rules.

tional cost. As a result, pruning becomes unstable, and heuristics that work well in continuous diffusion often fail in the discrete masked setting. Figure 3 illustrates this effect on the CoLA reward (Warstadt et al., 2019), where reward values fluctuate widely across denoising steps. Additional results for toxicity and sentiment rewards are provided in Appendix E, highlighting the generality of this challenge and motivating the need for alternative pruning strategies.

### 3.2 BRANCHING VIA UNMASKING FOR INCREASED BRANCH DIVERSITY

Recall that MDLMs pose unique branching difficulties: (i) parallel updates couple all masked tokens, so repeated resampling produces highly similar candidates; and (ii) the unmasking schedule is endogenous, so perturbing probabilities rarely alters which token is committed next.

To resolve these challenges, we branch only at commitment events. Using first-hitting sampling (Zheng et al., 2024), from the current unmasking time $\tau_n$, we jump directly to the next unmasking time $\tau_{n-1}$, evaluate the model once, and generate child nodes by (i) uniformly selecting a

masked index and (ii) enumerating the top-$b(n)$ tokens at that index (see Algorithm 1 for the pseudocode). This design introduces diversity in both unmasking *order* and token *content*, avoids wasted updates, and requires only one model call per parent node. The beam width $b(n)$ flexibly controls exploration, enabling wide yet efficient search at test time.

In comparison, naïve parallel sampling must repeatedly simulate transitions until unmasking occurs. We show in Theorem 1 that to obtain $b(n)$ child nodes from a parent node with $n$ masked tokens, it requires an average of $b(n)/(1 - \exp(-nh))$ model evaluations, where $h \ll 1$ is the discretization step. This cost grows rapidly as $h \to 0$, which is often necessary for accurate sampling (Sahoo et al., 2024). In contrast, our method performs only one model evaluation per parent node, yielding substantial speedup in branching.

---

**Algorithm 1** UNMASKBRANCH

**Require:** Parent node $(\boldsymbol{z}_n^{1:L}, \tau_n)$ with $n$ masks; beam width $b(n)$
1: $u_n \sim \mathcal{U}(0,1)$
2: $\tau_{n-1} \leftarrow \alpha^{-1}\big(1 - u_n^{1/n}(1 - \alpha_{\tau_n})\big)$
3: $\boldsymbol{\mu}_n \leftarrow \boldsymbol{x}_\theta(\boldsymbol{z}_n^{1:L}, \tau_{n-1})$
4: $\ell \sim \text{Unif}\{j \colon \boldsymbol{z}_n^{(j)} = \boldsymbol{m}\}$
5: $\mathcal{Z} \leftarrow \text{TopK}_{b(n)}(\boldsymbol{\mu}_n^\ell)$  $\qquad \triangleright \mathcal{Z}$ stores all the selected tokens
6: **return** $(\mathcal{Z}, \tau_{n-1}, \boldsymbol{\mu}_n)$

---

**Theorem 1 (Efficiency of UNMASKBRANCH over naïve parallel sampling)** *Fix a parent node with $n$ masked tokens and reparameterize time by $\gamma(t) = -\log(1 - \alpha_t)$. Discretize $\gamma$ on a uniform grid with step size $h \ll 1$. Run naïve parallel sampling (eq. (3)) and, in each run, stop at the first branch commitment (i.e., the first position that unmasks); repeat from the same parent node until $b(n)$ child nodes have been obtained. Then the expected total number of model evaluations required by the naïve parallel sampling is*

$$\mathbb{E}[\text{evals}] = \frac{b(n)}{1 - \exp(-nh)}. \tag{8}$$

*In contrast,* UNMASKBRANCH *produces $b(n)$ child nodes with exactly one evaluation. In particular, for $b(n) = 1$,* UNMASKBRANCH *matches the naïve-parallel first-change distribution over $(\tau_{n-1}, \ell)$, where $\tau_{n-1}$ is the next unmasking time and $\ell \in [n]$ is the index of the committed position.*

Beyond efficiency, we also consider the distributional behavior of UNMASKBRANCH in Theorem 1. For $b(n) = 1$, it produces the same distribution over the next unmasking time $\tau_{n-1}$ and committing index $\ell \in [n]$ as the first-change outcome of naïve parallel sampling, as originally shown by Zheng et al. (2024, Proposition 4.1). For completeness, we restate the result and provide an alternative proof in Appendix C. For $b(n) > 1$, however, naïve parallel sampling may unmask different positions across runs, whereas UNMASKBRANCH fixes a single index, leading to a different distribution.

### 3.3 PRUNING VIA RESUBSTITUTION FOR EFFICIENT REWARD EVALUATION

Pruning is equally challenging: (i) categorical predictions make sampled rewards noisy and unstable; and (ii) drawing extra completions per node inflates compute cost. As a result, naive scoring destabilizes search.

To address these challenges, we reuse the probabilities from branching to construct a provisional completion by *resubstitution*: committed tokens remain fixed, while masked positions are filled with current head predictions. This single

---

**Algorithm 2** RESUBSTITUTESCORE

**Require:** Candidate $\boldsymbol{z}_{n-1}^{1:L}$, probabilities $\boldsymbol{\mu}_n$
1: $\hat{\boldsymbol{x}}_0^{1:L} \leftarrow \boldsymbol{z}_{n-1}^{1:L}$
2: **for** masked position $\ell$ in $\boldsymbol{z}_{n-1}^{1:L}$ **do**
3: $\quad \hat{\boldsymbol{x}}_0^\ell \leftarrow \arg\max \boldsymbol{\mu}_n^\ell$  $\triangleright$ let $\hat{\boldsymbol{x}}_0^\ell$ be the token with the highest probability
4: **end for**
5: **return** $r(\hat{\boldsymbol{x}}_0^{1:L})$

---

proxy completion is scored once with the reward (see Algorithm 2 for the pseudocode). Resubstitution enables low-variance, deterministic scoring without extra model calls; it uses temporal locality by evaluating probabilities at the precise commitment time; and it ensures pruning costs to scale linearly with the number of parent nodes $m(n)$. Together, these properties enable stable, reward-aware pruning under the same NFE budget as baseline sampling.

We provide a theoretical justification for this pruning rule. Assuming the reward is Hamming–Lipschitz (Assumption 1), we show in Theorem 2 that the gap between the resubstituted reward and the true expected reward is bounded by the model's predictive uncertainty at masked positions.

**Assumption 1 (Hamming–Lipschitz reward)** *We assume that the reward function is Lipschitz continuous with respect to the Hamming distance, i.e., there exists a constant $\beta > 0$, such that for all $\boldsymbol{x}^{1:L}, \boldsymbol{y}^{1:L} \in \mathcal{V}^L$, we have*

$$\left| r(\boldsymbol{x}^{1:L}) - r(\boldsymbol{y}^{1:L}) \right| \leq \beta \cdot d_{\mathrm{H}}(\boldsymbol{x}^{1:L}, \boldsymbol{y}^{1:L}), \tag{9}$$

*where the Hamming distance is given by $d_{\mathrm{H}}(\boldsymbol{x}^{1:L}, \boldsymbol{y}^{1:L}) \coloneqq \sum_{\ell=1}^{L} \mathbf{1}_{\{\boldsymbol{x}^\ell \neq \boldsymbol{y}^\ell\}}$.*

**Theorem 2 (Resubstitution gap controlled by max confidence)** *Let $(\boldsymbol{z}_{n-1}^{1:L}, \tau_{n-1})$ be a state, and let $\boldsymbol{\mu}_n = \boldsymbol{x}_\theta(\boldsymbol{z}_{n-1}^{1:L}, \tau_{n-1})$ denote the model probabilities. Denote by $\mathcal{I}_{n-1} = \{\ell \colon \boldsymbol{z}_{n-1}^\ell = \boldsymbol{m}\}$ the set of masked indices. Let $\hat{\boldsymbol{x}}_0^{1:L}$ be the resubstituted completion from [Algorithm 2](#), and let $\boldsymbol{X}^{1:L}$ be a random completion obtained by sampling each masked index $\ell \in \mathcal{I}_{n-1}$ as $X^\ell \sim \mathrm{Cat}(\cdot; \boldsymbol{\mu}_n^\ell)$. Under [Assumption 1](#) and let $\beta$ be the Lipschitz constant therein, we have*

$$\left| \mathbb{E}\left[ r(\boldsymbol{X}^{1:L}) \right] - r(\hat{\boldsymbol{x}}_0^{1:L}) \right| \leq \beta \sum_{\ell \in \mathcal{I}_{n-1}} \left( 1 - \max_{v \in [V]} \boldsymbol{\mu}_n^\ell(v) \right). \tag{10}$$

The bound in [Theorem 2](#) justifies using resubstitution for pruning, as it shows that the approximation error is directly bounded by predictive uncertainty: when the model assigns high confidence to its top prediction, the resubstitution reward is close to the expected reward.

### 3.4 FULL TREASURE ALGORITHM

Having introduced the two key building blocks UNMASKBRANCH ([Algorithm 1](#)) for branching and RESUBSTITUTESCORE ([Algorithm 2](#)) for pruning, we now describe how TREASURE integrates them into a complete tree-search procedure, summarized in [Algorithm 3](#). Moreover, [Theorem 3](#) guarantees that increasing the tree width $m(n)$ always improves the final reward.

**Theorem 3 (Reward monotonicity in tree width)** *Fix the beam width $b(\cdot)$ and run TREASURE ([Algorithm 3](#)) twice with tree-width schedules $m(\cdot)$ and $m'(\cdot)$ such that $m'(n) \geq m(n)$ for all $n \in \{1, \ldots, L\}$. Couple all randomness across the two runs (same UNMASKBRANCH draws and model outputs), and use the same deterministic tie-breaking in $\mathrm{TopK}$. Let the returned rewards be $r_\star(m)$ and $r_\star(m')$. Then $r_\star(m') \geq r_\star(m)$.*

[Theorem 3](#) shows that increasing the tree width $m(n)$, which determines the number of model evaluations, leads to improved final rewards. This trend is verified empirically in [Section 4](#). As a remark, increasing the beam width $b(n)$ yields stronger local candidates, but this does not guarantee monotonic improvement, since locally better branches do not necessarily lead to higher final rewards.

---

**Algorithm 3** Tree Reward-Aligned Search with Unmasking and Resubstitution (TREASURE)

---

**Require:** Pretrained MDLM $\boldsymbol{x}_\theta(\boldsymbol{z}^{1:L}, t)$; reward $r(\cdot)$; length $L$; beam width $b(\cdot)$; tree width $m(\cdot)$
**Ensure:** Final sequence $\boldsymbol{z}_\star^{1:L}$ and reward $r_\star$
1: $\tau_L \leftarrow 1, \quad \boldsymbol{z}_L \leftarrow [\boldsymbol{m}, \ldots, \boldsymbol{m}]$
2: $\mathcal{S} \leftarrow \{(\boldsymbol{z}_L^{1:L}, \tau_L)\}$
3: **for** $n = L$ **down to** 1 **do**
4: $\quad \mathcal{C} \leftarrow \varnothing$                          ▷ candidate pool
5: $\quad$ **for all** $(\boldsymbol{z}_n^{1:L}, \tau_n) \in \mathcal{S}$ **do**
6: $\quad\quad (\mathcal{Z}, \tau_{n-1}, \boldsymbol{\mu}_n) \leftarrow$ UNMASKBRANCH$(\boldsymbol{z}_n^{1:L}, \tau_n, b(n))$      ▷ call [Algorithm 1](#)
7: $\quad\quad$ **for all** $\boldsymbol{z}_{n-1}^{1:L} \in \mathcal{Z}$ **do**
8: $\quad\quad\quad r \leftarrow$ RESUBSTITUTESCORE$(\boldsymbol{z}_{n-1}^{1:L}, \boldsymbol{\mu}_n)$           ▷ call [Algorithm 2](#)
9: $\quad\quad\quad \mathcal{C} \leftarrow \mathcal{C} \cup \{(\boldsymbol{z}_{n-1}^{1:L}, \tau_{n-1}, r)\}$
10: $\quad\quad$ **end for**
11: $\quad$ **end for**
12: $\quad \mathcal{S} \leftarrow \mathrm{TopK}_{m(n)}(\mathcal{C})$                  ▷ keep best $m(n)$ nodes
13: **end for**
14: $(\boldsymbol{z}_\star^{1:L}, \tau_0, r_\star) \leftarrow \arg\max_{(\boldsymbol{z}_0^{1:L}, \tau_0, r) \in \mathcal{S}} r$
15: **return** $\boldsymbol{z}_\star^{1:L}, r_\star$

---

## 4 EXPERIMENTS

In this section, we evaluate the performance of TREASURE on TTA for controllable text generation. All experiments are run on a single NVIDIA A100 GPU. Further experimental settings and additional results are provided in Appendix D and Appendix E.

### 4.1 CONTROLLABLE TEXT GENERATION

**Experimental settings.** We adopt the MDLM implementation by Sahoo et al. (2024) as our base model. To ensure a fair comparison across different TTA approaches, we follow the experimental protocol described in Han et al. (2023); Singhal et al. (2025) and fix the number of denoising steps to 1,000 for all the baseline methods. For each controllable prompt, we generate 20 continuations of length 128 tokens using 15 prompts, consistent with prior work. We report the NFE, defined as the total number of forward passes through the pretrained MDLM during generation, as the primary measure of test-time compute cost. Following prior work, we ignore the computational cost of the reward model, as it is shared across all methods and does not affect relative efficiency comparisons. More implementation details and hyperparameter configurations are provided in Appendix D.

**Baselines.** To evaluate the effectiveness of TREASURE, we compare it against several representative TTA approaches:

- **Base model sampling:** This baseline directly generates candidate outputs from a pre-trained MDLM without applying any additional optimization.

- **Best-of-$N$ (BoN):** This approach samples $N$ candidate sequences from the same pre-trained model and selects the one achieving the highest reward.

- **Feynman–Kac (FK) steering (Singhal et al., 2025):** A Sequential Monte Carlo (SMC)-based approach that maintains particles during generation and resamples them by reward-weighted importance, thereby improving sample efficiency and alignment.

**Rewards and metrics.** For controllable text generation, we consider four downstream reward functions: **(i) Perplexity** (GEN. PPL): Computed using GPT-2 (Radford et al., 2019), this metric encourages generations that are more likely under a pretrained language model. **(ii) Linguistic acceptability** (COLA): Based on a classifier (Morris et al., 2020) trained on the CoLA dataset (Warstadt et al., 2019), this reward favors sentences that are grammatically well-formed. **(iii) Toxicity score** (TOXICITY): Using a toxicity detection classifier (Logacheva et al., 2022) trained to identify harmful or offensive content, this reward assesses model vulnerabilities and penalizes toxic outputs. **(iv) Sentiment score** (SENTIMENT): Leveraging a sentiment classifier (Barbieri et al., 2020) trained on social media data, this reward guides the model toward producing outputs with the desired sentiment (e.g., positive). We evaluate model performance using the aforementioned reward functions as evaluation metrics, and additionally measure diversity (deferred to Appendix E) for a comprehensive assessment.

### 4.2 EXPERIMENTAL RESULTS

**Comparison with baseline methods.** We measure compute in terms of NFE. For BoN and FK-steering, following prior work, we fix the denoising steps at $T = 1,000$ and vary the per-step NFE across $\{1, 2, 4, 6, 8, 16\}$, leading to total budgets of $1,000 \times \{1, 2, 4, 6, 8, 16\}$ evaluations. Note that in TREASURE, the number of denoising steps naturally scales with sequence length (128 tokens in our setting), so total budgets are smaller but directly comparable in terms of per-step NFE. Table 1 reports results on controllable text generation across four reward functions: COLA, TOXICITY, SENTIMENT, and GEN. PPL. TREASURE consistently achieves state-of-the-art performance across all tasks. In the low-NFE regime, where compute is scarce, it already surpasses the best baseline results obtained at much higher NFEs. As NFE increases, performance improves steadily, as predicted by Theorem 3, and TREASURE dominates all baselines at every budget.

**Visualization of reward trajectories.** To understand how TTA interacts with MDLM denoising under TREASURE, we plot reward trajectories across denoising steps from 10 independent trials,

| METHOD | NFE | CoLA ↑ | TOXICITY ↑ | SENTIMENT ↑ | GEN. PPL ↓ |
|---|---|---|---|---|---|
| MDLMs | 1 | 25.56 | 0.89 | 12.44 | 80.58 |
| BoN | 2 | 43.16 | 0.70 | 22.11 | 70.58 |
| | 4 | 63.57 | 2.44 | 32.89 | 55.47 |
| | 6 | 65.96 | 2.57 | 45.50 | 52.23 |
| | 8 | 73.11 | 6.67 | 48.44 | 47.91 |
| | 16 | 77.56 | 11.11 | 65.56 | 42.46 |
| FK-steering | 2 | 45.96 | 1.05 | 20.00 | 66.79 |
| | 4 | 66.62 | 1.56 | 36.33 | 56.36 |
| | 6 | 69.82 | 3.16 | 41.05 | 50.16 |
| | 8 | 72.67 | 4.00 | 49.33 | 46.60 |
| | 16 | 76.44 | 9.67 | 61.33 | 41.56 |
| TREASURE | 2 | 77.67 | 64.00 | 98.67 | 15.37 |
| | 4 | 84.22 | 93.33 | 98.90 | 9.22 |
| | 6 | 89.19 | 96.60 | 99.11 | 7.60 |
| | 8 | 93.33 | 100.00 | 100.00 | 6.60 |
| | 16 | **98.35** | **100.00** | **100.00** | **5.11** |

Table 1: **Main results on TTA for controllable text generation with MDLMs.** We compare the base MDLM, Best-of-$N$ (BoN), FK-steering, and TREASURE (ours) across four reward functions: CoLA, TOXICITY, SENTIMENT, and GEN. PPL. TREASURE consistently outperforms all baselines, with especially large gains in low-NFE regimes and continued improvements as NFE increases. Arrows indicate whether higher (↑) or lower (↓) values are preferred. We remark that while lower TOXICITY is generally more desirable, increasing it also serves as a valid TTA target for benchmarking purposes.

with each colored curve corresponding to one trial. As shown in Fig. 4, rewards rise steadily during denoising. This reveals two properties of MDLM generation. First, the trajectory exhibits progressive refinement: early steps reconstruct coarse structures, while later ones refine them into coherent text. TREASURE exploits this by branching on alternative continuations and pruning low-reward ones. Second, the near-monotonic reward increase indicates that TREASURE not only improves final outputs but also steers intermediate states toward more desirable regions of the hypothesis space. Overall, TREASURE provides a stable mechanism for aligning MDLM outputs with reward signals throughout the denoising process.

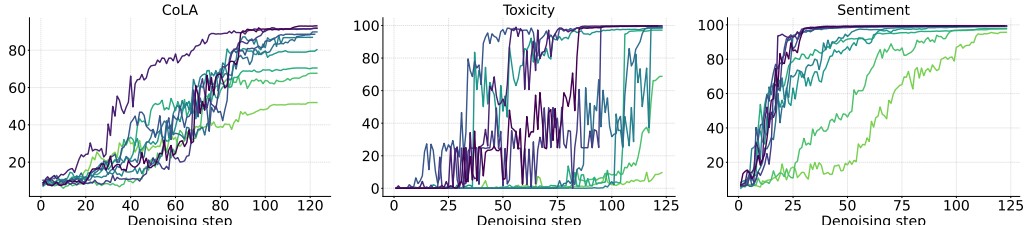

Figure 4: **Reward trajectories across denoising steps over** 10 **independent trials.** We show the evolution of rewards for three task-specific objectives (CoLA, TOXICITY, and SENTIMENT) (colors are sorted by final reward, but do not indicate temporal order). All three exhibit a generally increasing trend, illustrating progressive refinement during denoising and supporting the effectiveness of TREASURE. We omit GEN. PPL, as perplexity is computed before the first <EOS> token and thus depends on variable sentence lengths, making step-wise comparisons inconsistent.

### 4.3 ABLATION STUDIES

**Effectiveness of RESUBSTITUTESCORE.** To validate the effectiveness of deterministic scoring in Algorithm 2, where the masked positions in $z_{n-1}^{1:L}$ are replaced by the $\arg\max$ tokens of $x_\theta(z_n^{1:L}, \tau_{n-1})$, we compare against two natural alternatives: (i) *previous-step scoring*, which approximates the score using the posterior at the previous step, i.e., $r(z_n^{1:L})$. This corresponds to the

| METHOD | NFE | CoLA ↑ | TOXICITY ↑ | SENTIMENT ↑ | GEN. PPL ↓ |
|---|---|---|---|---|---|
| TREASURE | 2 | 77.67 | 64.00 | 98.67 | 15.37 |
| | 4 | 84.22 | 93.33 | 98.90 | 9.22 |
| | 6 | 89.19 | 96.60 | 99.11 | 7.60 |
| *previous-step scoring* | 2 | 45.01 | 26.67 | 93.33 | 60.14 |
| | 4 | 74.44 | 68.89 | 95.10 | 20.81 |
| | 6 | 73.33 | 73.94 | 97.27 | 20.03 |
| *true posterior scoring* | 10 ($= 5 \times 2$) | 84.33 | 93.33 | 98.79 | 10.09 |
| | 20 ($= 5 \times 4$) | 93.33 | 100.00 | 100.00 | 7.30 |
| | 48 ($= 8 \times 6$) | 93.33 | 100.00 | 100.00 | 5.11 |

Table 2: **Ablation study on RESUBSTITUTESCORE.** We compare the full TREASURE model (gray rows, copied from Table 1) against two variants: (i) *previous-step scoring*, which reuses the reward from the prior step, and (ii) *true posterior scoring*, which evaluates the exact posterior but inflates NFE (e.g., $10 = 5$ (beam width) $\times 2$ (tree width/NFE)). RESUBSTITUTESCORE achieves the best trade-off, outperforming (i) under matched NFE and matching (ii) with far fewer evaluations.

approach of Singhal et al. (2025), but ignores information from the current step and yields high-variance estimates; and (ii) *true posterior scoring*, which estimates the score from the true posterior $r(z_{n-1}^{1:L})$ as in Chung et al. (2023). While unbiased, this method requires additional NFEs per child node and thus incurs significant computational overhead. As shown in Table 2, RESUBSTI-TUTESCORE consistently outperforms the first approach under matched NFE, and achieves performance comparable to the second approach while requiring substantially fewer evaluations.

**Effectiveness of beam width.** We further study the role of beam width while keeping the tree width fixed at 4, so that the per-step NFE remains constant. Unlike tree width (Theorem 3), larger beam width does not guarantee better rewards. As shown in Figure 5, the reward may fluctuate as beam width grows, highlighting that beam search alone cannot reliably ensure better alignment under fixed tree width.

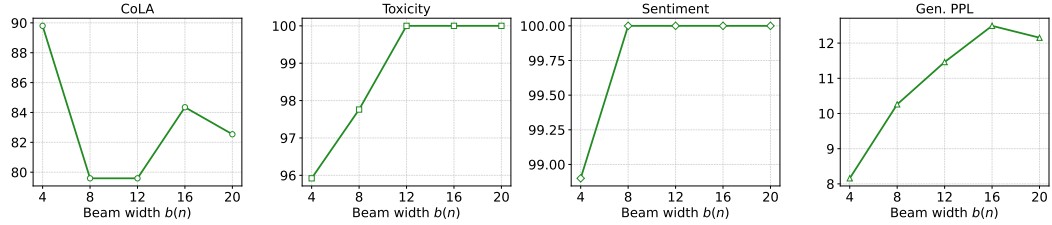

Figure 5: **Effect of beam width on reward.** We fix the tree width to 4, keeping per-step NFE constant, and vary the beam width. In contrast to tree width (cf. Theorem 3), increasing beam width does not guarantee better rewards, and performance may plateau or even degrade.

## 5 CONCLUSION

In this work, we proposed TREASURE, a tree-search method for test-time alignment in masked diffusion language models. It addresses the two central challenges for tree search in this setting: low-diversity branching under naïve parallel sampling and high-variance reward estimates due to distributional output. By introducing UNMASKBRANCH, which branches only at first-hitting unmask events, and RESUBSTITUTESCORE, which prunes via deterministic resubstitution, TREASURE achieves stable and compute-efficient alignment. Our analysis characterizes why these strategies are reliable, and our experiments demonstrate strong gains across diverse controllable generation tasks. Future work includes extending the approach to long-context and multimodal MDLMs, and developing theoretically-grounded adaptive schedules that better balance exploration and efficiency.

ETHICS STATEMENT

Our study uses only publicly available datasets and does not involve human participants. We are aware of potential misuse of machine learning models and emphasize that our contributions should be applied in a safe, fair, and responsible manner.

REPRODUCIBILITY STATEMENT

The code for implementing TREASURE will be released publicly upon publication. Theoretical results are stated with assumptions specified and complete proofs provided in Appendix C. Datasets are standard public benchmarks, and details of preprocessing and experimental settings are included in Appendix D.

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

**Roadmap.**   The appendix is organized as follows:

## A  ADDITIONAL LITERATURE REVIEW

Recent progress in large-scale language modeling spans three key directions: (i) Autoregressive Language Models (ARMs) remain dominant but suffer from sequential generation constraints; (ii) Masked Diffusion Language Models (MDLMs) offer a parallelizable alternative via coarse-to-fine refinement; and (iii) Test-Time Alignment (TTA) methods aim to steer model outputs toward desired objectives without fine-tuning. We briefly review each direction below.

**Autoregressive Language Models (ARMs).**   ARMs have achieved unprecedented success in the era of large-scale language modeling, powering cutting-edge systems such as ChatGPT (Achiam et al., 2023), DeepSeek (Guo et al., 2025a), and the Qwen series (Yang et al., 2025a), and driving significant advances toward Artificial General Intelligence (AGI). Following a causal next-token prediction paradigm, ARMs exhibit strong scaling properties and demonstrate impressive capabilities in reasoning (Shao et al., 2024; OpenAI, 2024), planning (Driess et al., 2023), and multimodal understanding (Liu et al., 2023; Wang et al., 2024; Chen et al., 2024b). However, their strict causal dependency introduces inherent limitations: generation is inherently sequential, inference remains computationally expensive, and controlling global properties such as structure and reasoning steps is challenging. This raises a natural question: *Is AR the only viable paradigm for achieving AGI?* Recently, an increasing number of studies have explored Masked Diffusion Language Models (MDLMs) (Sahoo et al., 2024) as an alternative framework, leveraging coarse-to-fine refinement and parallel decoding to rethink the foundations of large-scale language modeling.

**Masked Diffusion Language Models (MDLMs).**   Building on ideas from continuous diffusion models (Gao & Li, 2024; 2025; Ho et al., 2020), MDLMs stem from discrete diffusion models (Austin et al., 2021; Campbell et al., 2022; Lou et al., 2024; Sahoo et al., 2024; Shi et al., 2024) and demonstrate strong potential as an alternative to autoregressive paradigms. Closed-source systems such as Gemini Diffusion (DeepMind, 2024) and Mercury (Labs et al., 2025) achieve thousands of tokens per second, offering 5–10× faster generation than AR models of comparable size, highlighting the scalability and efficiency of the diffusion paradigm. On the open-source side, LLaDA (Nie et al., 2025) represents the first billion-scale MDLM trained from scratch on 2.3T tokens, achieving performance competitive with LLaMA-3-8B (Dubey et al., 2024) across reasoning, coding, and comprehension benchmarks. Building upon this, LLaDA-1.5 (Zhu et al., 2025) integrates reinforcement learning for preference alignment, further improving mathematical and code reasoning. In parallel, a continual pre-training paradigm adapts existing ARMs into MDLMs, with models such as DiffuLLaMA (Gong et al., 2024) or named DiffuGPT, and Dream-7B (Ye et al., 2025) demonstrating strong performance by leveraging pretrained backbones like LLaMA (Touvron et al., 2023) and Qwen2.5-7B (Team, 2024) while benefiting from the diffusion-native coarse-to-fine refinement process. This growing trend highlights MDLMs as a promising yet underexplored paradigm that enables parallel decoding, controllable refinement, and scalable training, motivating our investigation.

**Test-Time Alignment (TTA).**   Existing TTA methods can be broadly categorized into two families: (i) sampling-based approaches, which guide generation by adjusting the sampling distribution, and (ii) search-based strategies, which explicitly explore multiple decoding trajectories to identify high-reward outputs. Among sampling-based methods, the most straightforward is Best-of-$N$ (BoN) (Stiennon et al., 2020; Tang et al., 2024), a model-agnostic strategy that generates multiple candidates and selects the one achieving the highest reward. However, BoN is costly for diffusion models due to their iterative denoising. Recent work addresses this by integrating particle sampling

into the generation process, allowing multiple candidates to be explored in a single run. For instance, SVDD (Li et al., 2024) proposes selecting the highest-reward particle at every denoising step, while CoDe (Singh et al., 2025) extends this idea by performing selection only at specific intervals, effectively balancing computational efficiency and sample diversity. Generalizing the particle sampling paradigm, Sequential Monte Carlo (SMC) methods (Wu et al., 2023; Cardoso et al., 2023; Kim et al., 2025; Dou & Song, 2024) adopt a principled probabilistic framework that maintains a population of particles and iteratively performs importance weighting, resampling, and proposal optimization. Beyond sampling-based strategies, recent work has investigated search-based decoding strategies, including tree search (Li et al., 2025; Zhang et al., 2025; Ramesh & Mardani, 2025) and Monte Carlo Tree Search (MCTS) (Xie et al., 2024; Chen et al., 2024a; Zhang et al., 2024; Zhou et al., 2023), to improve alignment and reasoning performance by systematically exploring a larger set of candidate trajectories. Although sampling-based approaches (Singhal et al., 2025; Pani et al., 2025; Dang et al., 2025) have made early inroads into TTA for MDLMs, search-based methods remain underexplored. They have shown strong results in ARMs and continuous diffusion models. Extending them to MDLMs is still an open problem, which motivates this work.

## B  MORE DETAILS ON MDLMS

For completeness, we provide additional technical details underlying MDLMs used in the main text. Specifically, we (i) derive the form of the reverse process used during sampling, (ii) present the training objective and its connection to the negative Evidence Lower Bound (negative ELBO), and (iii) describe the deterministic constraints imposed on the denoiser outputs.

**Derivation of the reverse process.**   The reverse process follows Bayes' rule

$$q(\boldsymbol{z}_s|\boldsymbol{z}_t,\boldsymbol{x}) \propto q(\boldsymbol{z}_t|\boldsymbol{z}_s)q(\boldsymbol{z}_s|\boldsymbol{x}) \tag{11}$$

for $s < t$. By conditional independence across positions (Sahoo et al., 2024; Shi et al., 2024), it suffices to consider a single token. If $\boldsymbol{z}_t \neq \boldsymbol{m}$, then by the absorbing property of the mask token $\boldsymbol{m}$ in the forward process, the token cannot have been masked at time $s$. Consequently, the reverse transition is deterministic and simply copies back:

$$q(\boldsymbol{z}_s|\boldsymbol{z}_t,\boldsymbol{x}) = \delta_{\boldsymbol{z}_s,\boldsymbol{z}_t}. \tag{12}$$

If $\boldsymbol{z}_t = \boldsymbol{m}$, only $\boldsymbol{z}_s \in \{\boldsymbol{m}, \boldsymbol{x}\}$ have support. Using the forward transition $q(\boldsymbol{z}_t = \boldsymbol{m}|\boldsymbol{z}_s = \boldsymbol{m}) = 1$ and $q(\boldsymbol{z}_t = \boldsymbol{m}|\boldsymbol{z}_s = \boldsymbol{x}) = 1 - \alpha_t/\alpha_s$, together with the prior $q(\boldsymbol{z}_s = \boldsymbol{m}|\boldsymbol{x}) = 1 - \alpha_s$ and $q(\boldsymbol{z}_s = \boldsymbol{x}|\boldsymbol{x}) = \alpha_s$, the unnormalized weights are $w_{\boldsymbol{m}} = (1 - \alpha_s)$ and $w_{\boldsymbol{x}} = \alpha_s - \alpha_t$, with normalization $Z = w_{\boldsymbol{m}} + w_{\boldsymbol{x}} = 1 - \alpha_t$. Hence

$$q(\boldsymbol{z}_s|\boldsymbol{z}_t = \boldsymbol{m},\boldsymbol{x}) = \text{Cat}\Big(\boldsymbol{z}_s; \frac{(1-\alpha_s)\boldsymbol{m} + (\alpha_s - \alpha_t)\boldsymbol{x}}{1 - \alpha_t}\Big), \tag{13}$$

which is the expression used in eq. (2). Intuitively, the mass not yet absorbed by time $s$ splits between staying masked and reverting to the clean token $\boldsymbol{x}$, in proportions $(1 - \alpha_s)$ and $(\alpha_s - \alpha_t)$, respectively.

**Training and connection to the negative ELBO.**   MDLMs are trained by minimizing the negative ELBO, which decomposes into a data term and a diffusion term that fits the learned reverse kernel $p_\theta(\boldsymbol{z}_s|\boldsymbol{z}_t)$ to the true reverse dynamics $q(\boldsymbol{z}_s|\boldsymbol{z}_t,\boldsymbol{x})$ (Sahoo et al., 2024; Shi et al., 2024). In the discrete-time setting with $T$ steps, taking $s(i) = (i - 1)/T$ and $t(i) = i/T$, the diffusion term simplifies to a time-weighted masked cross-entropy:

$$\mathcal{L}_{\text{diff}} = \sum_{i=1}^{T} \mathbb{E}_{\boldsymbol{x},\boldsymbol{z}_{t(i)}} \left[ \frac{\alpha_{t(i)} - \alpha_{s(i)}}{1 - \alpha_{t(i)}} \big(-\log\langle \boldsymbol{x}_\theta(\boldsymbol{z}_{t(i)}, t(i)), \boldsymbol{x}\rangle\big) \right]. \tag{14}$$

Taking $T \to \infty$ yields the continuous-time negative ELBO (with $\alpha'_t$ the derivative of $\alpha_t$)

$$\mathcal{L}_\infty = \int_0^1 \frac{\alpha'_t}{1 - \alpha_t} \mathbb{E}_{\boldsymbol{x},\boldsymbol{z}_t \sim q(\cdot|\boldsymbol{x})}\big[-\log\langle \boldsymbol{x}_\theta(\boldsymbol{z}_t, t), \boldsymbol{x}\rangle\big]\,\mathrm{d}t. \tag{15}$$

For sequences with token-wise independent corruption and a factorized decoder (eq. (3)), this becomes a weighted average of MDLM losses over masked positions:

$$\mathcal{L}_\infty = \int_0^1 \frac{\alpha'_t}{1 - \alpha_t} \mathbb{E}\left[\sum_{\ell=1}^L \mathbf{1}_{\{\boldsymbol{z}_t^\ell = \boldsymbol{m}\}}\left(-\log\langle \boldsymbol{x}_\theta^\ell(\boldsymbol{z}_t^{1:L}, t), \boldsymbol{x}^\ell\rangle\right)\right]\mathrm{d}t. \tag{16}$$

Thus the continuous-time training loss is exactly the negative ELBO specialized to masking, penalizing reconstruction errors only at positions that are masked by the forward process.

**Denoiser output constraints.** We impose two output-time constraints on $\boldsymbol{x}_\theta$ via deterministic post-processing (i.e., by substitution rather than learning): (i) *Zero masking probabilities*, by setting the mask logit to $-\infty$, so $\langle \boldsymbol{x}_\theta(\boldsymbol{z}_t, t), \boldsymbol{m}\rangle = 0$; (ii) *Carry-over unmasking*, i.e., for any position already unmasked, copy through the observed value so that if $\boldsymbol{z}_t \neq \boldsymbol{m}$ then $\boldsymbol{x}_\theta(\boldsymbol{z}_t, t) = \boldsymbol{z}_t$. These constraints reflect the absorbing dynamics and ensure that only masked positions contribute to the loss, tightening the bound and stabilizing training and sampling.

## C    THEORETICAL GUARANTEE OF TREASURE

**Theorem 1 (Efficiency of UNMASKBRANCH over naïve parallel sampling)** *Fix a parent node with $n$ masked tokens and reparameterize time by $\gamma(t) = -\log(1 - \alpha_t)$. Discretize $\gamma$ on a uniform grid with step size $h \ll 1$. Run naïve parallel sampling (eq. (3)) and, in each run, stop at the first branch commitment (i.e., the first position that unmasks); repeat from the same parent node until $b(n)$ child nodes have been obtained. Then the expected total number of model evaluations required by the naïve parallel sampling is*

$$\mathbb{E}[\text{evals}] = \frac{b(n)}{1 - \exp(-nh)}. \tag{8}$$

*In contrast, UNMASKBRANCH produces $b(n)$ child nodes with exactly one evaluation. In particular, for $b(n) = 1$, UNMASKBRANCH matches the naïve-parallel first-change distribution over $(\tau_{n-1}, \ell)$, where $\tau_{n-1}$ is the next unmasking time and $\ell \in [n]$ is the index of the committed position.*

*Proof.* Let $n$ denote the number of masked positions at the parent node, and let $\tau_n$ be the current time. Reparameterize time by $\gamma(t) = -\log(1 - \alpha_t)$, and discretize $\gamma$ with uniform step size $h$. Suppose we run naïve parallel sampling and terminate as soon as any position unmasks.

Consider any masked position $\ell$. For $s' < s$ such that $\gamma_{s'} = \gamma_s - h$, the exact reverse transition gives

$$\mathbb{P}\left(\boldsymbol{z}_{s'}^\ell = \boldsymbol{m} \mid \boldsymbol{z}_s^\ell = \boldsymbol{m}, \boldsymbol{x}_\theta\right) = \frac{1 - \alpha_{s'}}{1 - \alpha_s} = \exp(-(\gamma_{s'} - \gamma_s)) = \exp(-h) \tag{17}$$

This means that the probability of the token $\boldsymbol{z}_s^\ell$ remaining masked is $\exp(-h)$, and the probability it unmasks is $1 - \exp(-h)$. Under naïve parallel sampling, the reverse transition is factorized as

$$p_\theta(\boldsymbol{z}_{s'}^{1:L} \mid \boldsymbol{z}_s^{1:L}) = \prod_{\ell=1}^L q(\boldsymbol{z}_{s'}^\ell \mid \boldsymbol{z}_s^{1:L}, \boldsymbol{x}_\theta), \tag{18}$$

so the transitions of masked tokens are conditionally independent. Therefore, for the $n$ masked tokens, the probability that no token unmasks in one step is

$$(\exp(-h))^n = \exp(-nh). \tag{19}$$

Thus, the probability that at least one token unmasks in a step is $p_h := 1 - \exp(-nh)$.

Let $G_h$ be the number of steps until the first unmasking occurs. Then for any $k \geq 0$,

$$\mathbb{P}(G_h > k) = (\exp(-nh))^k = \exp(-nhk). \tag{20}$$

Since the run halts at the first unmasking step, $G_h$ is geometric with success probability $p_h = 1 - \exp(-nh)$ on $\{1, 2, \dots\}$. Hence

$$\mathbb{E}[G_h] = \frac{1}{p_h} = \frac{1}{1 - \exp(-nh)}. \tag{21}$$

Now suppose we wish to obtain $b(n)$ child nodes by restarting from the same parent node and stopping each run at its first unmasking. Ignoring collisions (so that $b(n)$ runs yield $b(n)$ distinct children), the total number of model evaluations is $\sum_{r=1}^{b(n)} G_h^{(r)}$ with $G_h^{(r)}$ i.i.d. as $G_h$. Therefore,

$$\mathbb{E}[\text{evals}] = \sum_{r=1}^{b(n)} \mathbb{E}[G_h^{(r)}] = \frac{b(n)}{1 - \exp(-nh)}. \tag{22}$$

When $b(n) = 1$, the $\gamma$-time waiting $T_h := hG_h$ satisfies

$$\mathbb{P}(T_h > s) = \big(\exp(-nh)\big)^{\lfloor s/h \rfloor} \to \exp(-ns) \quad \text{as } h \to 0, \tag{23}$$

so $T_h$ converges in distribution to Exponential$(n)$. By the inverse-CDF representation, if $u \sim \mathcal{U}(0,1)$ then

$$\Delta_n := -\frac{1}{n} \log u \sim \text{Exponential}(n), \tag{24}$$

and hence

$$\gamma(\tau_{n-1}) = \gamma(\tau_n) + \Delta_n. \tag{25}$$

Mapping back via $\gamma(t) = -\log(1 - \alpha_t)$ yields the FHS draw

$$\tau_{n-1} = \alpha^{-1}\big(1 - u^{1/n}(1 - \alpha_{\tau_n})\big), \quad u \sim \mathcal{U}(0,1), \tag{26}$$

and by exchangeability the committing index is uniform on the $n$ masked positions. Thus for $b(n) = 1$ the naïve-parallel first-change law over $(\tau_{n-1}, \ell)$ coincides with FHS, while the expected number of model evaluations is $1/\big(1 - \exp(-nh)\big)$ for the grid scheme versus a single evaluation for UNMASKBRANCH. $\square$

**Theorem 2 (Resubstitution gap controlled by max confidence)** *Let $(z_{n-1}^{1:L}, \tau_{n-1})$ be a state, and let $\boldsymbol{\mu}_n = \boldsymbol{x}_\theta(z_{n-1}^{1:L}, \tau_{n-1})$ denote the model probabilities. Denote by $\mathcal{I}_{n-1} = \{\ell : z_{n-1}^\ell = \boldsymbol{m}\}$ the set of masked indices. Let $\hat{\boldsymbol{x}}_0^{1:L}$ be the resubstituted completion from Algorithm 2, and let $\boldsymbol{X}^{1:L}$ be a random completion obtained by sampling each masked index $\ell \in \mathcal{I}_{n-1}$ as $X^\ell \sim \text{Cat}(\cdot; \boldsymbol{\mu}_n^\ell)$. Under Assumption 1 and let $\beta$ be the Lipschitz constant therein, we have*

$$\left| \mathbb{E}\big[r(\boldsymbol{X}^{1:L})\big] - r\big(\hat{\boldsymbol{x}}_0^{1:L}\big) \right| \leq \beta \sum_{\ell \in \mathcal{I}_{n-1}} \left(1 - \max_{v \in [V]} \boldsymbol{\mu}_n^\ell(v)\right). \tag{10}$$

*Proof.* By Assumption 1, for any realization $\boldsymbol{X}^{1:L}$,

$$|r(\boldsymbol{X}^{1:L}) - r(\hat{\boldsymbol{x}}_0^{1:L})| \leq \beta d_{\mathrm{H}}(\boldsymbol{X}^{1:L}, \hat{\boldsymbol{x}}_0^{1:L}) = \beta \sum_{\ell=1}^{L} \mathbf{1}_{\{X^\ell \neq \hat{\boldsymbol{x}}_0^\ell\}}. \tag{27}$$

Taking expectations on both sides and applying the triangle inequality,

$$\left| \mathbb{E}[r(\boldsymbol{X}^{1:L})] - r(\hat{\boldsymbol{x}}_0^{1:L}) \right| \leq \beta \sum_{\ell=1}^{L} \mathbb{P}(X^\ell \neq \hat{\boldsymbol{x}}_0^\ell). \tag{28}$$

For already unmasked positions $\ell \notin \mathcal{I}_{n-1}$, we have $X^\ell = \hat{\boldsymbol{x}}_0^\ell$, so these terms vanish. For $\ell \in \mathcal{I}_{n-1}$, since $X^\ell \sim \text{Cat}(\cdot; \boldsymbol{\mu}_n^\ell)$ and $\hat{\boldsymbol{x}}_0^\ell = \arg\max_v \boldsymbol{\mu}_n^\ell(v)$,

$$\mathbb{P}(X^\ell \neq \hat{\boldsymbol{x}}_0^\ell) = 1 - \max_{v \in [V]} \boldsymbol{\mu}_n^\ell(v). \tag{29}$$

Substituting gives the bound in the statement. $\square$

**Theorem 3 (Reward monotonicity in tree width)** *Fix the beam width $b(\cdot)$ and run TREASURE (Algorithm 3) twice with tree-width schedules $m(\cdot)$ and $m'(\cdot)$ such that $m'(n) \geq m(n)$ for all $n \in \{1, \ldots, L\}$. Couple all randomness across the two runs (same UNMASKBRANCH draws and model outputs), and use the same deterministic tie-breaking in TopK. Let the returned rewards be $r_\star(m)$ and $r_\star(m')$. Then $r_\star(m') \geq r_\star(m)$.*

*Proof.* Write $\mathcal{S}_n(m)$ for the set of parent nodes after the $n$th expansion step when using schedule $m(\cdot)$. Thus at the start of the search we have $\mathcal{S}_L(m) = \{(\boldsymbol{z}_L^{1:L}, \tau_L)\}$, corresponding to the all-mask initial state, while at the end we obtain $\mathcal{S}_0(m)$, the collection of complete sequences with their rewards, from which the final $\arg\max$ is taken. Define $\mathcal{S}_n(m')$ analogously. Because the two runs are coupled (same UNMASKBRANCH uniforms, same committing indices, same model output $\boldsymbol{\mu}_n$), they produce the *same* candidate pool $\mathcal{C}_n$ at each level $n$.

At level $n$, selection applies $\text{TopK}_{m(n)}$ or $\text{TopK}_{m'(n)}$ to $\mathcal{C}_n$ with the same deterministic tie-breaking. Since $m'(n) \geq m(n)$, we have set inclusion

$$\text{TopK}_{m'(n)}(\mathcal{C}_n) \supseteq \text{TopK}_{m(n)}(\mathcal{C}_n), \tag{30}$$

and hence $\mathcal{S}_{n-1}(m') \supseteq \mathcal{S}_{n-1}(m)$. Inducting downwards from $n = L$ to $n = 1$ yields $\mathcal{S}_0(m') \supseteq \mathcal{S}_0(m)$.

The returned reward is the maximum of $r(\cdot)$ over the respective final sets:

$$r_\star(m) = \max_{(\boldsymbol{z}_0^{1:L}, \tau_0, r) \in \mathcal{S}_0(m)} r, \quad r_\star(m') = \max_{(\boldsymbol{z}_0^{1:L}, \tau_0, r) \in \mathcal{S}_0(m')} r. \tag{31}$$

Since a maximum over a superset cannot be smaller, $r_\star(m') \geq r_\star(m)$. The strictness condition follows immediately: if $\mathcal{S}_0(m') \setminus \mathcal{S}_0(m)$ contains a sequence with reward exceeding $\max_{\mathcal{S}_0(m)} r$, then the inequality is strict; otherwise the inequality reduces into an equality. $\square$

## D  EXPERIMENTAL DETAILS

**Details on beam width and tree width.**  To ensure reproducibility, we report two structural statistics of the search process:

- Beam width $b(n)$: At the $n$th step, the beam width denotes the number of child nodes expanded from a single parent node. This reflects the local branching factor of the search tree and controls how many alternatives are explored before pruning.

- Tree width $m(n)$. Defined as the number of nodes retained after pruning at $n$th expansion step. In other words, the tree width corresponds to the effective number of candidates that survive pruning, thereby characterizing the degree of parallel exploration. This quantity directly determines the number of function evaluations (NFEs) required at each step, since every surviving node must be expanded.

In all of our experiments, both the beam width and the tree width are set as fixed constants for each configuration. We denote each configuration in the format "$(b(n), m(n))$". For example, the settings $(5, 2), (6, 4), (8, 6), (12, 8), (20, 16)$ indicate that each parent node expands into 5, 6, 8, 12, or 20 branches, while the tree width is fixed to 2, 4, 6, 8, or 16, respectively.

**Details on baselines and evaluation metrics.**  All baseline results reported in this paper are obtained directly from the official implementation of FK-Diffusion-Steering (Singhal et al., 2025). All comparisons are based on running the released scripts without modification, unless otherwise specified. Similarly, all evaluation metrics are computed following the implementations provided in the same repository. We adopt the evaluation scripts released therein to guarantee fairness and consistency across methods. Specifically, the following pretrained models are used to calculate each metric:

- **Perplexity** (GEN. PPL): To encourage fluency during TTA, both the reward and the evaluation are based on generative perplexity computed with the pretrained GPT2-XL model (Radford et al., 2019).

- **Linguistic Acceptability** (CoLA): This metric favors grammatically well-formed sentences by employing a classifier (Morris et al., 2020) trained on the Corpus of Linguistic Acceptability (CoLA) (Warstadt et al., 2019). Importantly, CoLA classification accuracy is used consistently both as the reward signal during TTA and as the evaluation metric for reporting performance.

- **Toxicity** (TOXICITY): This metric leverages a toxicity detection classifier (Morris et al., 2020) to guide generation, with the task framed as red-teaming for harmful content. The goal is to assess

model vulnerabilities in producing toxic or offensive outputs. Consistently, toxicity classification accuracy (with "toxic" as the positive class) is used both as the reward signal during test-time alignment and as the evaluation metric for reporting performance.

- **Sentiment** (SENTIMENT): This metric employs a sentiment classifier (Barbieri et al., 2020) trained on social media text to guide outputs toward a desired polarity (e.g., positive sentiment). Consistently, sentiment classification accuracy (with "positive" as the target class) is used both as the reward signal during test-time alignment and as the evaluation metric for reporting performance.

## E  MORE EXPERIMENTS

### E.1  VARIATION OF REWARDS

**Experimental settings.**   To analyze the stability of reward signals during denoising, we generate full trajectories using first-hitting sampling (FHS). For each intermediate state $z_n^{1:L}$, we follow the reward estimation protocol of Singhal et al. (2025): we sample 1,000 candidate completions independently from the conditional distribution $p_\theta(\cdot|z_n^{1:L}, \tau_{n-1})$, compute the reward for each, and aggregate them to form an empirical distribution of reward estimates at that step. This allows us to track not only the mean reward but also its variability across the denoising process.

**Results.**   Figure 6 visualizes the distribution of estimated rewards at different denoising steps for two representative tasks, complementing Fig. 3. We observe that the reward estimates fluctuate substantially, with wide interquartile ranges and large min–max spans. This high variance highlights a key weakness of direct sampling-based estimation: small stochastic differences in sampled tokens can lead to disproportionately large changes in the measured reward. As a result, pruning decisions based on these noisy estimates can be unreliable, especially in the middle stages of denoising where uncertainty is greatest. These findings underscore the motivation for our proposed RESUB-STITUTESCORE, which provides low-variance, deterministic estimates and stabilizes pruning in tree search.

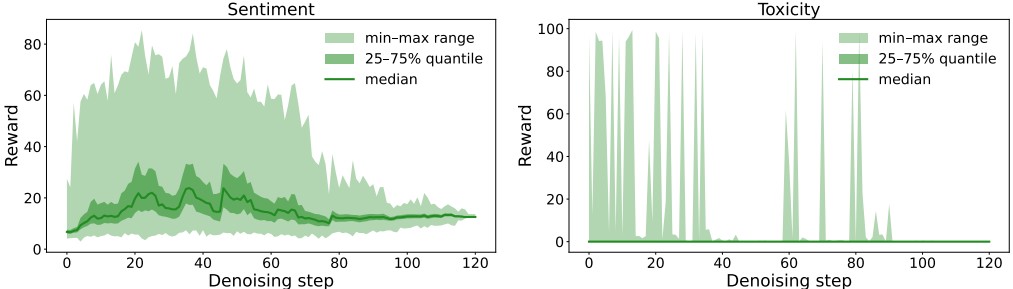

Figure 6: **Variation of direct reward estimates across denoising steps.** We estimate rewards by sampling 1,000 completions per step without applying TTA, following Singhal et al. (2025). Both SENTIMENT (left) and TOXICITY (right) exhibit high variance, with wide quantile bands and occasional outliers. In particular, TOXICITY shows large fluctuations: although unconditional generations are usually non-toxic, rare toxic samples in the estimation process can cause sharp spikes. These results illustrate the instability of direct sampling-based scoring and motivate the need for low-variance alternatives such as RESUBSTITUTESCORE.

### E.2  EFFECT OF INCREASING UNMASKING GROUPS

**Experimental settings.**   In addition to beam width $b(n)$ and tree width $m(n)$, we also examine the effect of increasing the number of *unmasking groups* in FHS. In the standard setting, FHS unmasks a single token uniformly from the remaining masked positions. Here, we generalize this by unmasking $k$ tokens simultaneously. For each of the $k$ positions, we compute the marginal distribution and select the top-$b(n)$ candidates, yielding $k \cdot b(n)$ child nodes. We then apply RESUBSTITUTESCORE to estimate their rewards and retain the top $m(n)$ nodes for expansion. We fix $b(n) = 5$ and $m(n) = 2$

across experiments, and vary $k \in \{1, 2, 3\}$ (with $k = 1$ corresponding to the standard TREASURE algorithm).

**Results.** Table 3 shows that increasing the number of unmasking groups consistently improves performance across all metrics. Larger groups expand the search space more aggressively, enabling FHS to find higher-reward continuations while keeping the pruning capacity $m(n)$ fixed. It is worth noting, however, that this strategy is a heuristic: unlike standard FHS, it no longer preserves the equivalence to naive parallel sampling. Nevertheless, the empirical gains suggest that parallel un-masking is a promising direction, and its theoretical implications merit further study.

| #UNMASKING GROUPS | CoLA ↑ | TOXICITY ↑ | SENTIMENT ↑ | GEN. PPL ↓ |
|---|---|---|---|---|
| 1 | 77.67 | 64.00 | 98.67 | 15.37 |
| 2 | 80.00 | 81.05 | 100.00 | 13.60 |
| 3 | 81.05 | 87.37 | 100.00 | 12.56 |

Table 3: **Effect of unmasking groups in FHS.** We vary the number of unmasking groups ($k$), which expands the child nodes per step from $b(n)$ to $k \cdot b(n)$. The gray row ($k = 1$) is directly copied from Table 1 as the baseline TREASURE. Increasing $k$ yields consistently higher rewards and lower perplexity, indicating that parallel unmasking can improve search quality without additional per-step NFEs. Although this heuristic breaks the formal equivalence between FHS and naive parallel sampling, it nonetheless provides a simple and effective enhancement.

### E.3 DIVERSITY MEASURE

We further report the diversity of generated samples under different reward functions, as an extension of the main results in Table 1. Following standard practice, we adopt Distinct-$n$ (DIST-$n$) metrics, including DIST-1, DIST-2, and DIST-3, which compute the ratio of unique $n$-grams to the total number of generated $n$-grams. These metrics capture the lexical diversity of the outputs: higher values indicate a greater variety of tokens and reduced repetition. All experimental settings remain identical to those described in the main text. As shown in Table 4, our method achieves the best overall performance while incurring only a marginal decrease in diversity.

## F USAGE OF LARGE LANGUAGE MODELS

We used Large Language Models (LLMs) to polish the writing, generate the cartoon explorer in Fig. 1, and assist with routine plotting code.

| METHOD (REWARD) | NFE | DIST-1 ↑ | DIST-2 ↑ | DIST-3 ↑ |
|---|---|---|---|---|
| MDLMs | 1 | 63.60 | 92.63 | 94.00 |
| BoN (CoLA) | 2 | 57.17 | 90.46 | 93.53 |
| BoN (TOXICITY) | 2 | 57.50 | 90.53 | 93.51 |
| BoN (SENTIMENT) | 2 | 56.99 | 90.75 | 93.85 |
| BoN (GEN. PPL) | 2 | 55.95 | 89.62 | 93.46 |
| BoN (CoLA) | 4 | 63.57 | 92.04 | 93.62 |
| BoN (TOXICITY) | 4 | 62.90 | 92.10 | 93.77 |
| BoN (SENTIMENT) | 4 | 62.50 | 92.58 | 94.11 |
| BoN (GEN. PPL) | 4 | 60.09 | 90.24 | 93.18 |
| BoN (CoLA) | 6 | 59.15 | 91.23 | 93.72 |
| BoN (TOXICITY) | 6 | 57.64 | 90.45 | 93.24 |
| BoN (SENTIMENT) | 6 | 57.09 | 91.50 | 94.20 |
| BoN (GEN. PPL) | 6 | 54.83 | 88.29 | 92.26 |
| BoN (CoLA) | 8 | 63.34 | 92.26 | 93.75 |
| BoN (TOXICITY) | 8 | 83.41 | 94.53 | 93.69 |
| BoN (SENTIMENT) | 8 | 62.56 | 92.57 | 94.28 |
| BoN (GEN. PPL) | 8 | 40.44 | 59.59 | 89.77 |
| BoN (CoLA) | 16 | 63.03 | 92.52 | 93.95 |
| BoN (TOXICITY) | 16 | 83.56 | 94.43 | 93.57 |
| BoN (SENTIMENT) | 16 | 62.27 | 92.57 | 94.29 |
| BoN (GEN. PPL) | 16 | 42.44 | 59.28 | 88.49 |
| FK-steering (CoLA) | 2 | 57.73 | 90.75 | 93.73 |
| FK-steering (TOXICITY) | 2 | 57.03 | 89.89 | 93.29 |
| FK-steering (SENTIMENT) | 2 | 56.60 | 90.30 | 93.69 |
| FK-steering (GEN. PPL) | 2 | 55.82 | 89.63 | 93.58 |
| FK-steering (CoLA) | 4 | 63.65 | 92.11 | 93.86 |
| FK-steering (TOXICITY) | 4 | 62.46 | 91.46 | 93.37 |
| FK-steering (SENTIMENT) | 4 | 62.80 | 93.28 | 94.55 |
| FK-steering (GEN. PPL) | 4 | 59.99 | 90.57 | 93.41 |
| FK-steering (CoLA) | 6 | 58.87 | 90.98 | 93.54 |
| FK-steering (TOXICITY) | 6 | 56.82 | 89.78 | 92.89 |
| FK-steering (SENTIMENT) | 6 | 55.89 | 90.67 | 93.71 |
| FK-steering (GEN. PPL) | 6 | 55.27 | 88.22 | 92.47 |
| FK-steering (CoLA) | 8 | 64.06 | 92.50 | 93.96 |
| FK-steering (TOXICITY) | 8 | 63.09 | 91.86 | 93.20 |
| FK-steering (SENTIMENT) | 8 | 63.19 | 93.21 | 94.36 |
| FK-steering (GEN. PPL) | 8 | 59.40 | 89.30 | 92.34 |
| FK-steering (CoLA) | 16 | 64.11 | 92.56 | 93.91 |
| FK-steering (TOXICITY) | 16 | 64.21 | 92.59 | 93.43 |
| FK-steering (SENTIMENT) | 16 | 62.53 | 93.07 | 94.32 |
| FK-steering (GEN. PPL) | 16 | 58.29 | 87.93 | 91.54 |
| TREASURE (CoLA) | 2 | 69.12 | 94.40 | 93.68 |
| TREASURE (TOXICITY) | 2 | 63.80 | 94.49 | 94.23 |
| TREASURE (SENTIMENT) | 2 | 61.05 | 94.91 | 94.98 |
| TREASURE (GEN. PPL) | 2 | 43.41 | 78.00 | 88.76 |
| TREASURE (CoLA) | 4 | 70.41 | 92.11 | 91.82 |
| TREASURE (TOXICITY) | 4 | 64.59 | 93.67 | 93.17 |
| TREASURE (SENTIMENT) | 4 | 60.09 | 94.39 | 94.83 |
| TREASURE (GEN. PPL) | 4 | 55.95 | 89.62 | 93.46 |
| TREASURE (CoLA) | 6 | 57.73 | 90.75 | 93.73 |
| TREASURE (TOXICITY) | 6 | 57.50 | 90.53 | 93.51 |
| TREASURE (SENTIMENT) | 6 | 56.99 | 90.75 | 93.85 |
| TREASURE (GEN. PPL) | 6 | 57.64 | 90.45 | 93.24 |
| TREASURE (CoLA) | 8 | 78.05 | 92.13 | 89.69 |
| TREASURE (TOXICITY) | 8 | 78.72 | 94.55 | 91.89 |
| TREASURE (SENTIMENT) | 8 | 80.85 | 97.14 | 94.92 |
| TREASURE (GEN. PPL) | 8 | 46.67 | 90.45 | 93.24 |
| TREASURE (CoLA) | 16 | 89.06 | 93.43 | 88.07 |
| TREASURE (TOXICITY) | 16 | 74.54 | 95.76 | 94.59 |
| TREASURE (SENTIMENT) | 16 | 83.19 | 97.01 | 94.73 |
| TREASURE (GEN. PPL) | 16 | 32.95 | 52.72 | 61.77 |

Table 4: **Diversity comparison under different methods and reward functions.** Results are reported in terms of DIST-$n$ metrics (DIST-1/2/3).

