# OpenReview forum: "Tree Reward-Aligned Search for TReASURe in Masked Diffusion Language Models"
_ICLR.cc/2026/Conference — Submitted to ICLR 2026_

### Official Review · Reviewer_ki3g · 2025-10-28

**Soundness:** 3
**Presentation:** 1
**Contribution:** 3
**Rating:** 4
**Confidence:** 3

**Summary:**

The paper proposes ideas to improve the use of tree search for test time adaptation of masked diffusion language models (MDLM). The proposals are to may understanding based on the insights of Zheng et al (2024), which provides an alternative view of masked diffusion language models and that points to their inefficiencies. This paper suggests using first hitting unmasking (the time it takes for the first next token to be unmasked in an MDLM) in the search process. The idea is to choose a random token at the first hitting time and branch by top tokens. The value estimate at a node is based on most probable completions at the node. The paper shows that these choices leads to significant improvement over best of N and Feynman-Kac, both in terms of accuracy for inference-time adaptation and faithfulness to the base model in terms of perplexity.

**Strengths:**

The proposed ideas are simple, natural and seem to be effective in practice.

I found Algorithms quite useful in understanding the proposed idea.

**Weaknesses:**

Despite having a simple and useful idea, I found the presentation of the paper quite confusing. I went through the introduction without understanding what the paper is planning to do and finally understood the idea using the provided “algorithms”. To give an example, the paper repeatedly refers to issues with “pruning” starting with the abstract, but the term, to my understanding, is not used appropriately. Examples like “high-variance estimates, making pruning unstable” or “ For pruning, it employs resubstitution scoring, which deterministically fills masked positions” makes me wonder if they mean “value estimation” by “pruning”. There is no actual pruning happening to the search tree as far as I see.

There are terms such as “tree width” and “beam width” used without definition, also noting that tree width has a specific meaning in computer science which is different from its use case here. There are also some minor but still problematic uses of language – e.g., when the paper refers to “two simplifications” when moving swapping the expectation and logarithm in Eq (6), making the reader wonder what the second change is. A useful addition to help with clarity would be to discuss a naive search algorithm as a basic application of search to MDLMs, which could then be improved; by doing so the paper can better explain the issues with the “standard” approach and introduce the necessary terminology.

Beyond presentation, there have been several other recent papers that explore search with diffusion, and some of them also consider applications to masked diffusion language models. The paper doesn’t use any search baselines in its evaluations and does not discuss and contrast with these works.

Zhang, Xiangcheng, et al. "Inference-time scaling of diffusion models through classical search." arXiv preprint arXiv:2505.23614 (2025).

Tang, Sophia, et al. "TR2-D2: Tree Search Guided Trajectory-Aware Fine-Tuning for Discrete Diffusion." arXiv preprint arXiv:2509.25171 (2025).

Jain, Vineet, et al. "Diffusion Tree Sampling: Scalable inference-time alignment of diffusion models." arXiv preprint arXiv:2506.20701 (2025).

Yoon, Jaesik, et al. "Monte carlo tree diffusion for system 2 planning." arXiv preprint arXiv:2502.07202 (2025).

**Questions:**

Please see under "weaknesses".

---

### Official Review · Reviewer_dsVE · 2025-11-01

**Soundness:** 1
**Presentation:** 3
**Contribution:** 2
**Rating:** 2
**Confidence:** 4

**Summary:**

The paper proposes TREASURE, a test-time alignment method for masked diffusion language models. Two main components are: (i) UNMASKBRANCH, which expands the search tree only at first-hitting unmasking events and produces children by enumerating Top-K tokens at a chosen masked position using a single MDLM forward pass; and (ii) RESUBSTITUTESCORE, which scores partial sequences by deterministically filling remaining masks with argmax tokens and evaluating the reward on that proxy completion. The paper provides theoretical backing in the form of a branching-cost advantage (Theorem 1), a Lipschitz error bound for resubstitution (Theorem 2), and monotonic reward improvement with increasing tree width (Theorem 3). Experiments on MDLM with reward functions CoLA, toxicity, sentiment, and GPT-2 perplexity report large gains vs Best-of-N and FK-steering at similar per-step NFE settings.

**Strengths:**

- Clear and practical idea tailored to challenges faced during TTA of MDLMs. Using first-hitting sampling to reduce the cost of branching is a sensible and feasible approach.
- The paper presents the various subroutines clearly, and they seem fairly easy to implement. The overall writing is easy to understand.
- Theorem 1 clarifies where the branching-cost advantage comes from; Theorem 3 formalizes the intuitive role of tree width. Theorem 2 offers a qualitative stability proxy for resubstitution.

**Weaknesses:**

The main weaknesses of the paper are poor evaluation (does not check for over-optimization/reward-hacking), scalability of the approach (cost grows linearly with sequence length) and clarifications on certain statements in the paper.

- **Evaluation shows high rewards, but not overall quality and coherence.** My main concern is with evaluation. Since the goal is to sample from the reward-tilted posterior (Section 2.2), simply reporting high reward is not enough - it needs to be balanced with high likelihood under the model. In other words, I suspect the large gains in toxicity/CoLA/sentiment score or large PPL drops may be accompanied by a drop in overall coherency, quality, and faithfulness to the prompt, since these reward functions are notoriously bad evaluators [1,2] and can be hacked by simply repeating certain words. This is also corroborated with much lower diversity for TREASURE compared to BoN and FK-Steering for CoLA and PPL (dist-3 scores in the appendix). I suggest the authors (1) report evaluation using humans or LLM-as-judge, since it has been shown that they align better with human judgements [3], and (2) provide qualitative examples to show the method results in still coherent text that responds to the prompt appropriately.
- **Compute cost is linear in sequence length.** Due to the modified branching method, the number of steps for TREASURE grows linearly with the sequence length. The remaining baselines need a fixed number of denoising steps regardless of the output sequence length; indeed, this is one of the main strengths of diffusion LMs compared to autoregressive LMs. This raises concerns about the scalability of the method, and if the cost is growing linearly with sequence length, one may opt to use autoregressive LMs since they generally outperform diffusion LMs.
- **Variance across multiple seeds.** It seems all results are reported for a single seed. I *strongly* suggest the authors report results + variance for multiple seeds (and, where relevant, across prompts) to improve the reliability of the results.
- **Scope of Theorem 2 (resubstitution).** Theorem 2 bounds the gap between argmax resubstitution and the expected reward under independent one-shot token draws at the current step. It does not bound the gap to the *true reward* obtained by iterative unmasking with refreshed conditionals (i.e., the operational process during decoding). The claim that resubstitution yields “reliable intermediate reward estimates” is too strong without evidence that it is calibrated with respect to the true reward.
- **Assumption 1 realism (Hamming-Lipschitz reward).** For language rewards like sentiment/toxicity/CoLA, a single token flip can cause a discontinuous change (e.g., *happy*→*sad*). The Hamming-Lipschitz constant can be very large, making the bound loose or insignificant. This limitation should be acknowledged and, ideally, quantified empirically.
- **Diversity of UNMASKBRANCH vs naïve parallel sampling.** UNMASKBRANCH fixes an index and enumerates Top-K tokens at that index, which forces token-level diversity at one position, whereas naïve decoding allows unmasking different positions. I am not sure why UNMASKBRANCH is preferable here (apart from reducing computation), since this may reduce overall diversity/increase correlation in the children. The difference between the distributions is acknowledged in the paper, but since the motivation is to reduce early correlation, the paper should either analyze this diversity trade-off (index-diversity vs token-diversity), or explain that UNMASKBRANCH is purely for reducing computation and does not help solve the correlation/diversity problem.
- **Positioning vs prior work.** Diffusion Tree Sampling (DTS) [4] is a directly relevant tree-based method for continuous and discrete diffusion, ReMDM [5] proposes a remasking scheme for TTA of MDLMs, and SoP [6] proposes a search over path methods for continuous diffusion; these works should be discussed and contrasted The paper describes [7] as “continuous diffusion + tree search,” but [7] spans both continuous and discrete (MDLM) and is closer to beam search than a full tree expansion. Statements should be rephrased to accurately reflect these works.
- **Soft value approximation.**
    - The paper mentions in line 167 that one-step predictions in continuous diffusion lead to “reliable intermediate reward estimates”. This statement is not true, as observed in prior work [4], due to the high bias of these estimates. I request the authors to rephrase this statement.
    - When discussing soft value estimation for MDLMs, line 201 states that “sampling multiple completions … but doing so greatly increases computational cost”. But given the model’s prediction of the distribution at each token, merely sampling different sequences does not incur any further model calls and hence should be quite cheap. Could the authors clarify what leads to high computational cost here?
- **TREASURE seems like beam search rather than full tree search.** TREASURE maintains a frontier of top $M$ candidates, expands each into $K$ children and then prunes it back to $M$. This is exactly how a beam search operates, and there is no backtracking or dynamic tree traversal, unlike a classical tree search.

*[1] Holtzman, Ari, et al. "The Curious Case of Neural Text Degeneration." International Conference on Learning Representations, 2020.*

*[2] Hashimoto, Tatsunori B., Hugh Zhang, and Percy Liang. "Unifying Human and Statistical Evaluation for Natural Language Generation." Proceedings of the 2019 Conference of the North American Chapter of the Association for Computational Linguistics: Human Language Technologies, Volume 1 (Long and Short Papers). 2019.*

*[3] Liu, Yang, et al. "G-Eval: NLG Evaluation using Gpt-4 with Better Human Alignment." Proceedings of the 2023 Conference on Empirical Methods in Natural Language Processing. 2023.*

*[4] Jain, Vineet, et al. "Diffusion Tree Sampling: Scalable inference-time alignment of diffusion models." The Thirty-ninth Annual Conference on Neural Information Processing Systems, 2025.*

*[5] Wang, Guanghan, et al. "Remasking discrete diffusion models with inference-time scaling." arXiv preprint arXiv:2503.00307 (2025).*

*[6] Ma, Nanye, et al. "Inference-time scaling for diffusion models beyond scaling denoising steps." arXiv preprint arXiv:2501.09732 (2025).*

*[7] Li, Xiner, et al. "Dynamic Search for Inference-Time Alignment in Diffusion Models." arXiv preprint arXiv:2503.02039 (2025).*

**Questions:**

- The probability calculus in Theorem 1 is fine, but the *compute* conclusion implicitly assumes no sharing across $b(n)$ independent children when estimating the expected steps to first commit. A well-engineered “naïve parallel” baseline can share per-step logits across siblings until each hits its first commit, reducing total distinct model evaluations from $\sim b(n)/p$ to $\sim \log b(n)/p$, where $p = 1-\exp(-nh)$. TREASURE still enjoys a branching-cost advantage (one call for $K$ children), but the quantitative gap vs an amortized baseline is smaller than implied. Given the diversity trade-off (discussed in weaknesses), could the authors comment on the compute vs. diversity trade-off assuming a more fair naïve parallel cost model?
- For the FK-steering baseline, what potential function was used? What was the value of the temperature $\lambda$? The paper mentions they used “the running scripts as is”, but it is not very informative to a reader; please detail all hyperparameters and choices for the baselines.
- How exactly is the true posterior in Table 2 computed? The listed NFEs (e.g., 10=5×2; 20=5×4; 48=8×6) seem much smaller than what full iterative denoising would require to obtain faithful conditional updates per token. Are these partial rollouts, truncated schedules, or exact posteriors under additional assumptions?
- Are reward plots in Figure 4 computed on fully decoded sequences (not resubstitution proxies)? If so, please specify the decoding path from the tree to the final sequence.

---

### Official Review · Reviewer_eok7 · 2025-11-02

**Soundness:** 3
**Presentation:** 4
**Contribution:** 3
**Rating:** 4
**Confidence:** 4

**Summary:**

This paper addresses the problem of test-time reward alignment for masked diffusion language models (MDLMs). The authors identify two key challenges when applying tree search to MDLMs: (1) parallel unmasking produces highly correlated branches that limit exploration, and (2) reward evaluation through sampled completions yields high-variance estimates, making pruning unstable. To resolve these issues, the paper proposes new branching and pruning strategies.

UNMASKBRANCH improves exploration efficiency by applying first-hitting unmasking: from each parent node, it randomly selects one masked position to reveal and expands the top-k predicted tokens at that position as child nodes. This diversifies both token order and content while requiring only one model call per parent node. RESUBSTITUTESCORE stabilizes pruning by reusing predictions from branching to compute rewards deterministically. It fills remaining masked tokens with their top-1 predicted values to form a proxy completion, yielding a low-variance reward estimate without extra model evaluations.

Theoretical analysis demonstrates efficiency improvements and bounded approximation errors, while experiments show state-of-the-art performance across diverse rewards. TREASURE achieves consistently higher rewards than prior methods.

**Strengths:**

1. Method is well-motivated. The paper identifies issues in adapting tree search to masked diffusion language models and introduces simple and theoretically grounded solutions.
2. The paper is well-written and self-contained, with clear algorithms and well-illustrated diagrams.
3. Experimental results show consistent improvements over baselines.

**Weaknesses:**

1. The study focuses on masked diffusion language models, and including experiments on diverse backbones (e.g., LLaDA [1]) would further validate the generality of the proposed approach.
2. The work could be strengthened by incorporating additional baselines that, although not originally designed for MDLMs, can be easily applied to this setting (e.g., [2, 3]). Including such comparisons would make the empirical claims more compelling.

I would be willing to increase the score if the above concerns are adequately addressed.

[1] Nie et al., Large Language Diffusion Models, NeurIPS 2025.

[2] Ma et al., Inference-Time Scaling for Diffusion Models
beyond Scaling Denoising Steps, arXiv preprint.

[3] Li et al., Derivative-Free Guidance in Continuous and Discrete Diffusion Models with Soft Value-Based Decoding, arXiv preprint.

**Questions:**

Please refer to Weaknesses section.

---

### Meta-Review · Area_Chair_cRsN · 2026-01-06

**Summary:**

All reviewers agree to reject.

**Reviewer Concerns:**

No rebuttal.

**Reviewer Scores:**

No rebuttal.

---

### Decision · Program_Chairs · 2026-01-26

Reject